# Sequence variants associated with BMI affect disease risk through BMI itself

Gudmundur Einarsson [1], Gudmar Thorleifsson[1], Valgerdur Steinthorsdottir [1], Florian Zink[1], Hannes Helgason[1,2], Thorhildur Olafsdottir [1], Solvi Rognvaldsson[1], Vinicius Tragante [1], Magnus O. Ulfarsson[1,3], Gardar Sveinbjornsson [1], Audunn S. Snaebjarnarson [1], Hafsteinn Einarsson [1,2], Hildur M. Aegisdottir[1], Gudrun A. Jonsdottir[1], Anna Helgadottir [1], Solveig Gretarsdottir [1], Unnur Styrkarsdottir [1], Hannes K. Arnason[1], Ragnar Bjarnason[4,5], Emil Sigurdsson[6,7], David O. Arnar[1,4,8], Einar S. Bjornsson[4,9], Runolfur Palsson [4,9], Gyda Bjornsdottir [1], Hreinn Stefansson[1], Thorgeir Thorgeirsson[1], Patrick Sulem [1], Unnur Thorsteinsdottir[1,4], Hilma Holm [1], Daniel F. Gudbjartsson [1,2] & Kari Stefansson [1,4] ✉

Mendelian Randomization studies indicate that BMI contributes to various diseases, but it's unclear if this is entirely mediated by BMI itself. This study examines whether disease risk from BMI-associated sequence variants is mediated through BMI or other mechanisms, using data from Iceland and the UK Biobank. The associations of BMI genetic risk score with diseases like fatty liver disease, knee replacement, and glucose intolerance were fully attenuated when conditioned on BMI, and largely for type 2 diabetes, heart failure, myocardial infarction, atrial fibrillation, and hip replacement. Similar attenuation was observed for chronic kidney disease and stroke, though results varied. Findings were consistent across sexes, except for myocardial infarction. Residual effects may result from temporal BMI changes, pleiotropy, measurement error, non-linear relationships, non-collapsibility, or confounding. The attenuation extent of BMI genetic risk score on disease associations suggests the potential impact of reducing BMI on disease risk.

Body mass index (BMI) is a well-recognized risk factor for many chronic disorders[1], including type 2 diabetes mellitus[2] (T2D), several cardiovascular diseases[3–5] (CVDs), non-alcoholic fatty liver disease[6], chronic kidney disease[7] (CKD) and osteoarthritis[8]. Although the role of body mass in the pathogenesis of these diseases is not fully understood, the biomechanical forces imposed by higher BMI may contribute to some, such as knee osteoarthritis, which demonstrates a stronger association with obesity than hand osteoarthritis[8,9].

BMI is a highly polygenic trait. The largest genome-wide association study (GWAS) of BMI to date included 700 thousand individuals of European origin and yielded 941 sequence variants[10] that account for 6% of the population variance in BMI[10]. Sequence variants associated with BMI present an opportunity to investigate how and to what extent obesity confers risk of diseases[11–14], e.g., with Mendelian randomization (MR)[15]. Several studies have been performed that support a direct contribution of obesity to the pathogenesis of

[1]deCODE genetics/Amgen, Inc., Reykjavik 102, Iceland. [2]School of Engineering and Natural Sciences, University of Iceland, Reykjavik 102, Iceland. [3]Faculty of Electrical and Computer Engineering, University of Iceland, Reykjavik, Iceland. [4]Faculty of Medicine, University of Iceland, Reykjavik 102, Iceland. [5]Children's Medical Center, Landspítali University Hospital, Reykjavík, Iceland. [6]Development Centre for Primary Health Care in Iceland, Reykjavík, Iceland. [7]Department of Family Medicine, University of Iceland, Reykjavik, Iceland. [8]Cardiovascular Services, Landspitali University Hospital, Reykjavik, Iceland. [9]Internal Medicine Services, Landspitali University Hospital, Reykjavik, Iceland. ✉ e-mail: kari.stefansson@decode.is

diseases, such as CVD s[14], multiple other chronic diseases[16], and general all-cause mortality[17].

One of the assumptions of MR is that the instruments affect disease risk solely through exposure, in this case, BMI. Here, we wished to investigate whether disease risk conferred by sequence variants that are associated with BMI is mediated through their effects on BMI or by other mechanisms. To this end, we performed mediation analysis to explore the extent to which adult BMI mediates the risk conferred by BMI variants on various diseases, which, to our knowledge, has not been conducted before. We utilized genetic risk scores for BMI (BMI-GRSs) and compared the association between the GRSs and the various diseases with and without adjustment for measured BMI. The GRS in the main analysis had variants excluded that were deemed outliers in the BMI disease relationships, as these likely violate the MR assumption that they should affect disease risk solely through exposure. Any residual association between the BMI-GRSs and diseases after adjusting for measurements of adult BMI could be due to pleiotropy, confounding, or information not captured with a few adult BMI measurements. Assuming that the association of the BMI-GRS is mediated through BMI, the proportion mediated provides an estimate of the upper bound of the disease risk that is potentially modifiable by lowering the BMI.

## Results

### Genetic risk score associations

We used BMI measurements and genotype data from 139,236 Icelanders and 429,700 persons of European descent from the UK Biobank with a mean of 4 and 1.2 BMI measurements for each individual in the Icelandic and UK datasets, respectively. Of the 941 reported sequence variants that are associated with BMI[10], we used a subset of 665 independent variants to generate GRSs (pairwise $R^2 < 0.2$ in the UK). We furthermore excluded 45 variants that deviate from the typical pattern of association with BMI and disease risk for the 10 diseases we investigated (Fig. 1, Supplementary Data 1 and Supplementary Figs. 1–10). T2D had the highest number of outliers out of the ten diseases, or 25 of the 45 outlier variants. Figure 2 displays a scatter plot of the effects of the 665 BMI variants on BMI and T2D. Most BMI variants that associate with greater BMI correlate positively with T2D, but some have a strong negative correlation (Fig. 2)[18]. Most of the discordant variants were first identified through their association with T2D[19] and are likely to associate with BMI through T2D, a form of reverse causation.

We constructed two GRSs based on the selected variants to test for association with 10 diseases or conditions reflecting diseases in the Icelandic[20] and UK[21] datasets: atrial fibrillation, CKD, non-alcoholic fatty liver disease, glucose intolerance, heart failure, hip or knee replacement because of severe osteoarthritis, myocardial infarction (MI), stroke, and type 2 diabetes (Supplementary Data 1). The weights for the Icelandic GRS analysis were derived from a meta-analysis of BMI data from the UK and the 2015 GIANT GWAS[22] and that for the UK GRS analysis from a meta-analysis of recent Icelandic data and the 2015 GIANT data. The variance explained in the BMI phenotype by the BMI-GRSs was 4.9% in Iceland and 5.5% in the UK (Supplementary Table 1).

We scaled each GRS score such that a unit change corresponded to a predicted one unit (1 kg/m²) increase in BMI in the respective population (Supplementary Table 2). Thus, when we test the BMI-GRS for association with disease, the resulting odds ratios (ORs) can be interpreted as a change in disease risk per 1 kg/m² increase in BMI. The BMI-GRS was associated with all ten diseases in the Icelandic dataset and with all but stroke in the UK data ($p < 0.05/10$). The most significant association with BMI-GRS in both populations, and with the largest effect size, was with T2D ($OR_{Iceland} = 1.23$, $p_{Iceland} = 1.2e\text{-}128$, $OR_{UK} = 1.26$, $p_{UK} = 4.3e\text{-}448$). The associations with BMI-GRS and the diseases were mostly similar in the two populations (Supplementary Table 3), but we did observe significant differences for T2D ($p_{het} = 4.0e\text{-}3$), MI ($OR_{Iceland} = 1.06$, $OR_{UK} = 1.11$, $p_{het} = 7.1e\text{-}5$) and HF ($OR_{Iceland} = 1.10$, $OR_{UK} = 1.17$, $p_{het} = 7.2e\text{-}6$).

### Mediation analysis

We then performed mediation analysis by conditioning on measured BMI (Fig. 3, Supplementary Fig. 11 and Supplementary Table 3). The association of the BMI-GRS with knee replacement, glucose intolerance, and fatty liver disease was fully attenuated after conditioning on BMI in both the UK and Icelandic datasets (Fig. 3 and Supplementary Table 3). For the remaining diseases, the associations were substantially, but not fully, attenuated in both datasets, with the proportion of effect mediated ranging from 29% to 85%. This attenuation was mostly consistent in the two populations; the 95% confidence intervals for the proportion of effect mediated overlapped in the two populations for all diseases, except for CKD, 49% (95% CI: 33%–67%) attenuation in Icelandic data and 96% (95% CI: 85%–115%) attenuation in UK data.

A)

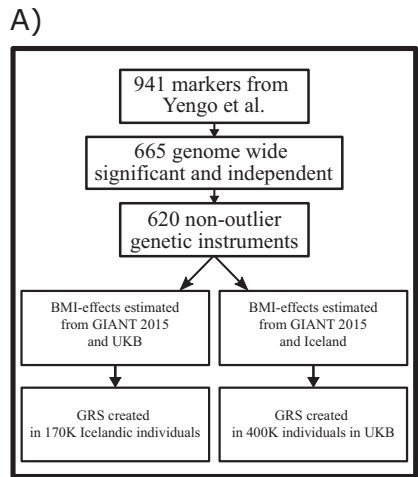

B)

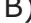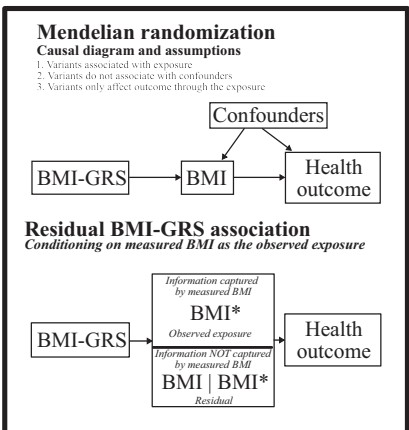

**Fig. 1 | Schematic overview of the key features of the study. A** depicts a diagram of how the BMI-GRSs were constructed. **B** depicts in the top section the causal diagram for Mendelian Randomization with the three assumptions. The lower part of (**B**) shows how the residual associations can be interpreted. In the middle, we have a box representing BMI as the exposure, in the upper part we have measured BMI, represented as BMI* and the lower part, we have BMI|BMI*, which represents the information not contained in the measurements. The information not contained in the measurements can have an impact on disease, demonstrated by significant residual associations.

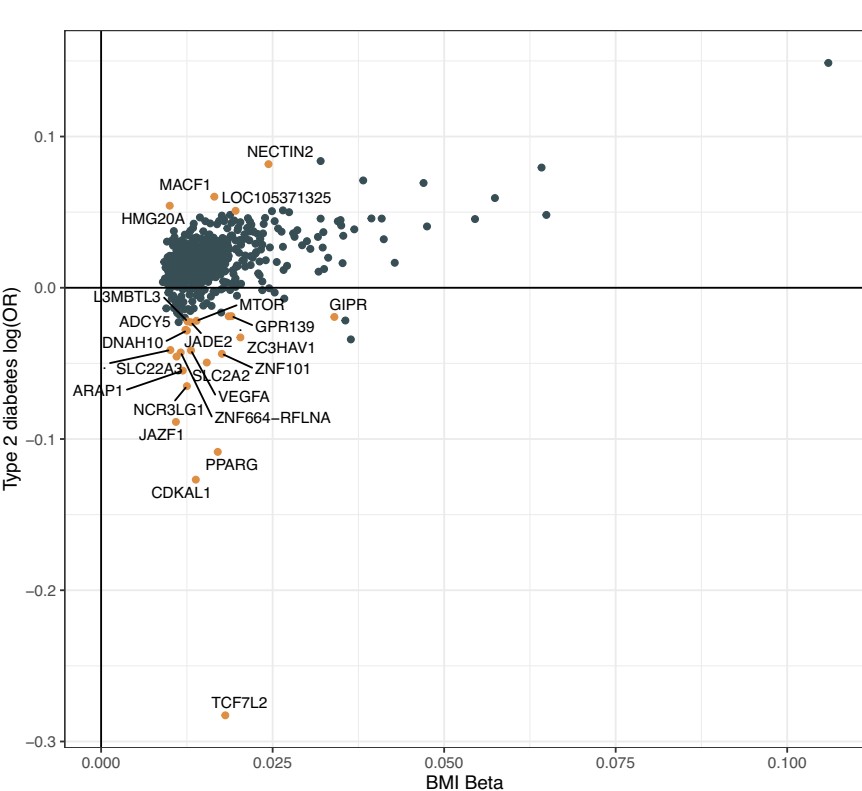

**Fig. 2 | Scatter plot showing effects of BMI variants on BMI on the *x* axis and their effects (log(OR)) on T2D on the *y* axis.** The log(OR) is meta-analyzed from Icelandic and UK data, while the BMI Beta is from Yengo et al.[10]. Outliers detected with MRPRESSO are colored in orange and annotated with the closest gene. A bidirectional relationship is observed, most of the variants are concordant, while most of the outliers are discordant and have strong associations with T2D. Source data are provided as a Source Data file.

## Sensitivity analysis with potential confounders

The associations that remain after conditioning on BMI could be explained by other variables that are associated with the BMI variants, BMI, and/or the diseases. To investigate this, we performed sensitivity analysis by adjusting for three additional variables of particular relevance to BMI and the diseases, namely ever-smoking status[23], waist-to-hip ratio[24] (WHR) and educational attainment[25] (EA), (Supplementary Note). Adding any one of the three variables, or all together, as covariates had little or no effect on the adjusted associations of the BMI-GRS with the diseases. We also included T2D as a covariate for all the associations except the one with T2D. Including T2D information did not have an effect on the adjusted association (See Supplementary Fig. 12).

## Correlations of repeated measurements

To assess the stability of measured adult BMI, we investigated repeated measures of BMI in Icelandic data (Supplementary Fig. 13). The estimated correlations for BMI measured 5 and 10 years apart for ages 18–80 years, are similar through most of adulthood, ranging from 0.7 to 0.9, until the age of 60–70 years, where the correlations start to decline (Supplementary Fig. 14). This demonstrates that through most of adult life, BMI trajectories follow a predictable path.

## GRS variance explained by age

Subsequently, in the UK dataset, we assessed how the BMI-GRS accounts for BMI variance across different age groups (Supplementary Fig. 14). We found that the BMI-GRS explained ~6% of BMI variance in individuals aged 40–55 years, decreasing slightly yet significantly to 5%

in those aged 65–75 years. This pattern implies an age-related shift in the BMI-GRS's predictive accuracy.

## Mediation analysis by sex

We performed sex-stratified mediation analysis in the UK data (Supplementary Fig. 15 and Supplementary Tables 4 and 5). Results were generally concordant between the sexes with respect to effect estimates, both conditioned on BMI and not. MI was the only disease where we observed differences in effects between sexes ($OR_{females} = 1.08$, $OR_{males} = 1.13$, $p = 0.0012$) in the unadjusted analysis. Furthermore, the BMI-GRS association was fully attenuated among females, but not males when conditioning on BMI ($OR_{males\ adj} = 1.05$, $p = 5.7e-10$). Thus, the residual association with MI in the joint analysis of the sexes seems to be driven by males.

## Sensitivity to outlier removal

To investigate the impact of removing outlier variants on the mediation analysis, we performed the same analysis with two less restrictive genetic scores, one including all genome-wide significant independent BMI variants, referred to as BMI-GRS-with-Outliers (Supplementary Tables 1 and 6), and another including less significant markers, the BMI polygenic risk score (BMI-PRS), created with 611 K markers using LDPred[26] (Supplementary Tables 1 and 7). The proportion of the GRS association mediated by BMI for the three different scores is summarized in Supplementary Table 8. We also computed effect estimates of the associations of the rank-transformed BMI phenotype to compare the effects of the genetic scores with an epidemiological estimate in the same population (Supplementary Table 9). As shown in Fig. 4,

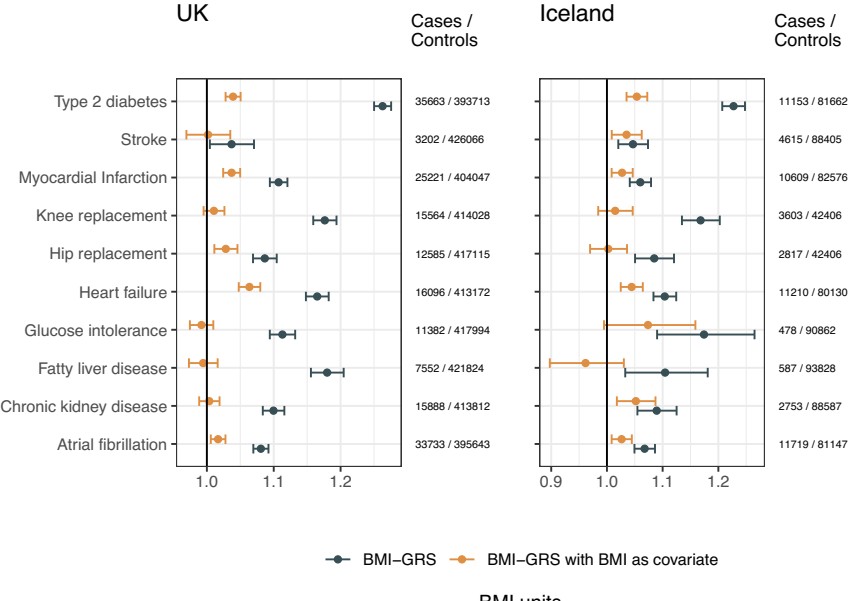

**Fig. 3 | Main association results summarized in a forest plot.** Shown are associations between the BMI-GRS and diseases/conditions in the UK Biobank (left) and Iceland (right). The GRSs were in both cases scaled such that an increase by one unit corresponded to a 1 kg/m² increase in BMI, thus the ORs correspond to a 1 kg/m² increase in BMI. The black points and error bars show the disease association of the BMI-GRS adjusted for sex, year of birth, and 20 genetic principal components), and the orange points and error bars show the corresponding association when BMI has been added as a covariate. The error bars correspond to a 95% confidence interval for the parameter, and the points represent ORs from logistic regression. See Supplementary Table 3 for details. Source data are provided as a Source Data file.

the effect estimates of the BMI-GRS and the BMI-PRS on T2D were different in both populations, and the BMI-GRS-with-Outliers effect was smaller than the BMI-GRS effect in the UK, but not in the Icelandic data. The BMI-PRS effects tended to be smaller than the BMI-GRS effect estimates indicative of the bias stemming from reverse causation variants, a similar pattern was observed for the BMI-GRS-with-Outliers, but to a lesser extent. The effect of the BMI-GRS on HF was greater than that of BMI on HF. This difference may partly be attributed to the nonlinear effects of BMI on HF risk (Supplementary Notes).

In the mediation analyses for glucose intolerance, for both BMI-PRS and BMI-GRS-with-Outliers, we observed a shift in the effect of associations: they changed from positive in the unadjusted analysis to negative upon adjusting for BMI (Supplementary Figs. 16–19 and Supplementary Tables 6 and 7). These differences also result in different estimates for the proportion of association mediated by BMI for the different scores (Supplementary Figs. 20 and 21, and Supplementary Table 8). The T2D association is fully erased when conditioned on measured BMI, both in the analysis with the BMI-GRS-with-Outliers and the BMI-PRS, but we observe a residual association with T2D in the BMI-GRS association. These results highlight the importance of removing outlier sequence variants from the analysis, as they violate the third MR assumption that the variants should only associate with the outcome through the exposure, and bias the results[15]. Analyzing the outlier variants exclusively further highlights these differences (Supplementary Note).

## Discussion
Here we show that the conservative BMI-GRS (with outlier variants excluded) associated with various diseases, atrial fibrillation, CKD, fatty liver disease, glucose intolerance, heart failure, osteoarthritic hip or knee replacement, MI, stroke, and type 2 diabetes, confirming previous MR results supporting a causal role of BMI in the pathogenesis of these diseases. We show that the associations of the BMI-GRS

with the diseases are largely attenuated when conditioned on measured adult BMI. This is consistent with BMI contributing directly to their development. However, there are disease-specific differences in the proportion of the genetic risk mediated through measured BMI that are not adjusted for by adult BMI measurements. For instance, measured BMI fully accounts for the BMI-GRS association with fatty liver disease and knee replacement, while it accounts for most but not all the association of BMI-GRS with T2D, MI, and HF.

The adult BMI measurements contain incomplete information about the total exposure of BMI. The incompleteness does not stem from measurement accuracy, but rather the limited temporal coverage of repeated measurements since BMI can change over time. Adult BMI follows a predictable path, yet a limited number of measurements cannot fully inform us about the lifetime BMI trajectory. One of the limitations of this study is that confounders can also influence the effect of the BMI-GRS when we condition on measured BMI, but it is likely that they would bias the effect further from the unadjusted effect towards zero (Supplementary Notes). Another limitation is that MR assumes a linear relationship between the exposure and outcome, nonlinear relationships can give rise to incomplete attenuation when conditioning on BMI, the putative exposure. Assuming that the BMI-GRS confers disease risk through BMI, the proportion of disease risk mediated in our mediation results provides an estimate of an upper bound of the genetic risk of the diseases that is potentially modifiable with a reduction in weight.

It is also possible that BMI itself is very correlated with the true causal factor for some of the diseases investigated, a form of vertical pleiotropy, and that BMI mediates this association. Evidence from a clinical trial of a GLP-1 agonist as a treatment for major CVDs demonstrates that the protective effect of treatment is observed from the start of intervention and is consistent throughout the trial period[27], yet the full effect of the treatment on weight is not observed until one year after treatment initiation[28]. This is not consistent with the effect on

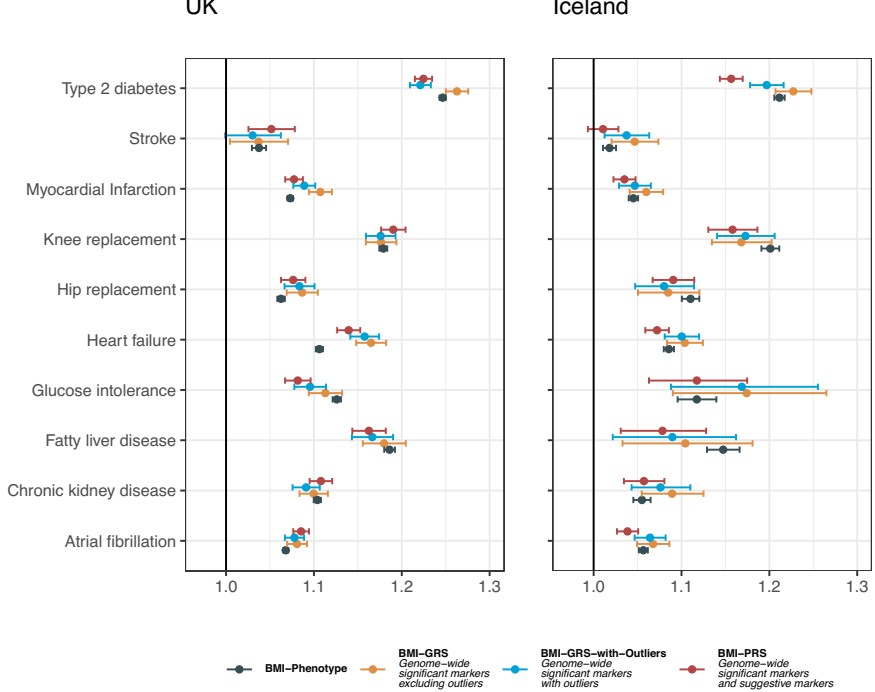

## Effect estimate comparison by risk scores

1 Unit change in BMI

**Fig. 4 | Comparing estimation of BMI as a risk factor for diseases by different approaches in UK Biobank and Icelandic data.** Three scores are compared, BMI-GRS (yellow), BMI-GRS-with-Outliers (blue), and the BMI-PRS (red), all created from BMI effects from BMI-GWAS meta-analyses (See methods and Fig. 1). All scores have been scaled such that a unit increase corresponds to a 1 kg/m² increase in BMI, thus the ORs correspond to a 1 kg/m² increase in BMI. The BMI-GRS is the same as the one reported in Fig. 3. The BMI phenotype corresponds to the black legend and represents an epidemiological estimate of the risk for the corresponding population. The counts of cases and controls are the same as reported in Fig. 3. The points and error bars show the disease association of the BMI genetic scores and the BMI phenotype adjusted for sex, year of birth, and 20 genetic principal components. The error bars correspond to a 95% confidence interval for the parameter, and the points represent ORs from logistic regression. See Supplementary Tables 3, 6, 7, and 9 for full details of the associations. Source data are provided as a Source Data file.

CVD being directly caused by BMI, since that would give rise to a dose-response relationship. BMI could be a consequence of an underlying causal mechanism, which could, for instance, be excessive food intake, adipocyte expansion, or adipogenesis. This pattern of cardiovascular event incidence curves has also been observed for bariatric surgery[29].

The effects of BMI-GRS on disease risk were generally larger than for BMI-GRS-with-Outliers and BMI-PRS, with the latter likely underestimating the causal effect due to the inclusion of reverse causality instruments or horizontal pleiotropy. While our BMI measurements capture much of the BMI-GRSs impact on disease risk, they are still incomplete. The significant residual association of the BMI-GRS with T2D, MI, and HF after adjusting for BMI calls for further research into these persisting correlations. Comprehensive BMI trajectory data could shed light on whether the residual associations are due to missing temporal information. On the other hand, the BMI-GRS might also be linked with other intermediary factors, such as food preferences or levels of physical activity. Presently, our results establish an upper bound on the disease risk directly attributed to measured BMI. Further research is required to determine the lasting risks associated with significant weight gain.

## Methods
### Study populations
**Iceland.** The Icelandic deCODE database contains a collection of data from Icelanders who have participated in several genetic initiatives. Our analysis was restricted to those aged over 18 years. All participants who donated biological samples signed informed consent for research. To ensure participant confidentiality, a sanctioned encryption system was employed and monitored by the Icelandic Data Protection Authority. The study was approved by the Icelandic National Bioethics Committee (approval No. VSN-17-076). The study complies with all relevant regulations regarding the use of data from human participants and was conducted in accordance with the criteria set by the Declaration of Helsinki. No statistical method was used to predetermine the sample size.

Height, weight and BMI information from Icelanders was obtained from Landspítali−The National University Hospital of Iceland; the Primary Health Care Clinics of the Capital area; and from a range of research undertakings at deCODE genetics. In case of multiple height measurements, the mode of the measurements was selected to represent the individual's height, thus a unique height measurement was obtained for each individual. BMI's were adjusted by regressing out the effect of age at measurement using a generalized additive model with splines from the mgcv package[30] separately for each sex. Multiple measurements were averaged after the adjustment process. The residuals from each sex were then inverse normal transformed.

Information about diseases was largely obtained from Landspítali −The National University Hospital and several registries managed by the Directorate of Health: Causes of Death Register, Register of Primary Health Care Contacts and Register of Contacts with Medical Specialists in Private Practice. Cases of atrial fibrillation were defined by ICD-10 code I48 (and sub-codes) and ICD-9 code 427.3. Cases of CKD were defined by ICD-10 code N18 (and sub-codes). Fatty liver disease was defined by ICD-10 code K76.0 (and sub-codes). The glucose intolerance diagnoses were based on ICD-10 code R73 (and sub-codes). The heart failure diagnoses were based on ICD-10 code I50

(and sub-codes). Hip or knee replacements because of osteoarthritis were based on NOMESCO Classification of Surgical Procedures (NCPS) codes NFB and prior ICD-10 M16.0, M16.1, or M16.9 codes in the same individual for hip replacement, or NGB and prior ICD-10 M17.0, M17.1, or M17.9 codes for knee replacement, or were obtained from the national Icelandic knee or arthroplasty registry[31]. Individuals who underwent joint replacement after the age of 40 years were included, and those with diagnosis of rheumatoid arthiritis were excluded. MI cases were based on ICD-10 codes I21 (and sub-codes) or I25.2 and comparable ICD-9 codes[32]. Ischemic stroke cases, were identified based on the ICD-9 codes 433,434 and ICD-10 codes I63.0, I63.1,I63.2, I63.3, I63.4, I63.5, I63.8, I63.9[33]. Type 2 diabetes cases were defined by ICD-10 code E11 or at least two measurements of hemoglobin A1c (HbA1C) > 6.5% or use of oral diabetes medication or self-report. Diagnosis of type 1 diabetes or MODY was used as an exclusion criterion. In general, for Icelandic case-control analyses, the controls were individuals recruited through different genetic studies at deCODE Genetics who had not been diagnosed with the disease in question.

**United Kingdom.** The UK Biobank (UKB) is a vast biomedical repository holding intricate phenotypic and genetic data on about 500,000 individuals from England, Wales, and Scotland. We restricted our analysis to individuals of European descent. They were between 40-69 years old at the point of entry. All participants willingly gave their informed consent. The operational methodologies were vetted and sanctioned by The North West Research Ethics Committee, and access to the UKB resources was gained through application no. #56270. The study complies with all relevant regulations regarding the use of data from human participants and was conducted in accordance with the criteria set by the Declaration of Helsinki. No statistical method was used to predetermine the sample size.

BMI was obtained from data field f.21001. The BMI measure was adjusted for year of birth, age, age squared, and 20 principal components for males and females separately, then combined, the average drawn from multiple measurements for individuals, and then inverse normal transformed. Data for the sexes was combined after the inverse normal transform.

Disease information was aggregated from multiple data fields, including hospital medical records (f.42170), primary clinical event records (f.42040), self-reported illness (f.20002), hospital records of surgical procedures (f.41149), self-reported medical conditions (f.2443) and death register (f.40001/40002). The atrial fibrillation diagnoses were based on ICD-10 code I48 (and sub-codes) from hospital medical records, primary clinical events records, or self-reported illness. CKD diagnoses were based on ICD-10 codes N18.3, N18.4, and N185 from hospital medical records or primary care clinical events records. Fatty liver disease diagnoses were defined by ICD-10 code K760 from hospital records or primary care clinical events records. Glucose intolerance was defined by ICD-10 code R73 (and sub-codes) from hospital medical records or primary care clinical events records. Heart failure diagnoses were defined by ICD-10 code I50 (and sub-codes) from hospital medical records, primary care clinical events records, or self-reported illness. Hip replacement was based on OPSC-4 codes, and matched ICD-10 codes, from hospital operation records. The OPCS-4 codes included are W371, W378, W379, W381, W388, W389, W391, W399 matched with ICD-10 codes M160, M161, M169 and with age at operation greater than 40 years. Similarly, knee replacement was based on OPSC-4 codes, and matched ICD-10 codes, from hospital records of surgical procedures. The OPCS-4 codes included are W401, W408, W409, W411, W418, W419, W421, W428, W429 matched with ICD-10 codes M170, M171, M179 and with age at surgery greater than 40 years. MI was defined using ICD-10 codes I200, I21, I210, I211, I21.2, I21.3, I21.4, I21.9, I21.X, I22, I22.0, I221, I22.8, I22.9, I25.2 from hospital medical records, primary care clinical events records or

from the death register. Cases of Ischemic stroke were defined using ICD-10 codes I63.0, I631, I63.2, I63.3, I63.4, I63.5, I63.8 from hospital medical records. Type 2 diabetes diagnoses were based on ICD-10 code E11 from hospital medical records, primary care clinical events records, self-reported illness, death register, and self-reported medical condition. Individuals with type 1 diabetes (ICD-10 E10) or gestational diabetes (ICD-10 O244) were excluded.

Quantitative traits were used for additional adjustments in the UK data. WHR is computed from waist circumference (field f.48) and hip circumference (field f.48). The values were adjusted in the same way as the BMI UK data. The smoking data used for adjustment was aggregated from the fields f.2897 Age stopped smoking, f.20116 Smoking status, and f.20161 Pack years of smoking. The years of education were determined from UK Biobank field 6138 and translated into specific durations using the following previously described[34] rules, 1) College or University degree corresponds to 20 years; 2) professional qualifications, such as nursing or teaching, equate to 18 years; 3) qualifications like A levels/AS levels, CSEs, NVQ, HND, or HNC are set at 13 years; 4) O levels/GCSEs or their equivalents are pegged at 10 years. 5)Individuals with none of the aforementioned qualifications are assigned 7 years. Subsequently, adjustments to this data were made in the same way as for the BMI data.

## Sequencing data

In both populations studied, genetic variants were first identified in whole-genome sequencing data. Subsequently, these variants were imputed into the remaining dataset based on chip-genotyping via long-range phasing[35]. Single nucleotide polymorphisms were called using Graphtyper[20,36,37].

**Iceland.** At deCODE genetics, 64,460 individuals had their whole genomes sequenced, while 173,025 underwent chip-genotyping as part of multiple research endeavors. The sequencing technologies employed included GAIIx, HiSeq, HiSeqX, and NovaSeq from Illumina.

**UK.** Within the UK Biobank initiative, all participants were chip-genotyped, and 131,958 participants had whole-genome sequencing performed using the NovaSeq Illumina devices[38].

## Method details

**BMI markers for genetic risk scores.** Starting with the list of 941 markers published by Yengo et al.[10], we first mapped the base-pair positions to hg38 and queried the markers in our data. We matched 905 markers that were used for subsequent analysis. We removed all markers with $p > 5e\text{-}8$, with the $p$ value which was computed in the original inverse variance weighted meta-analysis. After this step, we ended up with 673 markers. We then used LD data estimated from the UK Biobank to remove markers that had an $R^2$ of 0.2 or higher with the strongest variant in the region. This process yielded 665 markers that we refer to as the set of independent and genome-wide significant markers, (Fig. 1 and Supplementary Data 1).

**Genetic risk scores outlier removal.** Due to the existence of outliers in the associations, e.g., the *TCF7L2* variant for T2D (Fig. 2), we wanted to remove variants that strongly deviate from the typical association pattern, which could stem from horizontal pleiotropy or reverse causation. We, therefore, used the MRPRESSO method[39] to remove outliers. We can only know that we have removed the instruments that are most severely deviating from the expected distribution. Removing outlier variants is particularly important for the mediation analysis, as strong reverse causation variants can substantially bias the results.

**Meta-analysis for outlier estimation.** Using BMI effects estimated by Yengo et al.[10] for the exposure, we computed using inverse variance

meta-analysis the log ORs (ORs) for the 10 disease traits, using data from Iceland and UK Biobank. These log ORs were used as an outcome in the MRPRESSO analysis. Any outlier detected in the 10 MRPRESSO runs was removed from subsequent analysis. A variant was deemed as an outlier if the Bonferroni corrected outlier test was below 0.05. This required drawing 20 K samples from the null distribution to achieve enough precision to compute the empirical $p$-values. We removed 45 outlier markers from the subsequent analysis (See supplementary Figs. 1–10). For certain traits we found no outliers.

**Constructing GRSs.** When constructing the GRSs we made sure to use additive genetic effects estimated with the data that we had available, excluding the data from the target cohort. This is to ensure that we have no biases stemming from sample overlap or overfitting[40,41]. The effects were meta-analyzed from three summary statistics: (1) 2015 GIANT Consortium data (excluding the Icelandic data); (2) Icelandic data; and (3) UK Biobank data. The PRS score was created using the infinitesimal model, with all markers.

**GRS associations with diseases and mediation analysis.** The GRS associations and mediation analysis were performed using R version 3.6.3 and the lm and glm functions from the stats package[42]. The GRS was used as a predictor for the disease outcomes using logistic regression implemented in the glm function. In all associations, we included covariates for sex, year of birth, and 20 genetic principal components. In the mediation analysis, we simply added the measured BMI as a covariate to the model, that is, the inverse normal transformed adjusted BMI.

**GRS associations with BMI.** We computed the association of the GRSs, the GRSs-with-outliers, and the PRSs with the inverse normal transformed BMI phenotype. This was done using the lm function in R using the same covariates as for the disease association. The reported $p$ values are from a two-sided test. We also computed the effects with raw BMI data. In order to account for multiple measurements, we simply used the BMI at first visit in the UK data, while for the Icelandic data, we computed the mean of all BMI measures for each individual (Supplementary Table 2). These values were used to scale the effects of the GRS association with diseases, in order to interpret the ORs as a proportional increase in risk with respect to one BMI unit increase. This furthermore allowed us to compare the effects of the different scores in Fig. 4 in a common unit. We also computed this scale for each sex separately in the UK data (Supplementary Table 5).

**GRS associations adjusting for confounders.** In the mediation analysis, we added measurements for four potential confounders, namely EA, WHR, T2D, and smoking. The EA and WHR phenotypes were added as simple extra terms to the model. The smoking adjustment was done by including three terms, a factor of smoking status, and then a nonlinear function of pack years and years since the individual had stopped smoking (Supplementary Notes). The T2D adjustment was not done for the T2D association. The combined adjustment included WHR, EA, and smoking.

**Proportion-mediated confidence intervals.** In order to create confidence intervals for the proportion of effect attenuated (See Supplementary Table 8), we used the R package mediation[43], as has been previously been done for PRS scores[44].

## Software
We used R version 3.6.3 for the analysis and visualizations using the packages tidyverse (v1.3.0), ggsci (v2.9) and ggrepel (v0.8.2). Other software used was Ensembl version 87, https://www.ensembl.org/index.html Graphtyper version 2, https://github.com/DecodeGenetics/graphtyper, BOLT-LMM version 2.1, https://data.broadinstitute.org/alkesgroup/BOLT-LMM/downloads/, IMPUTE2 version 2.3.1, https://mathgen.stats.ox.ac.uk/impute/impute_v2.html, dbSNP version 140, http://www.ncbi.nlm.nih.gov/SNP/.

## Reporting summary
Further information on research design is available in the Nature Portfolio Reporting Summary linked to this article.

## Data availability
The SNPs used to construct the BMI-GRS scores can be found in the Supplementary Data 1 file along with information on which are classified as outliers. The Icelandic individual level data cannot be made publicly available due to data privacy laws. Those wishing to access the Icelandic individual level data should contact the corresponding author, Kari Stefansson (kari.stefansson@decode.is), to organize a visit to deCODE Genetics' facilities in Iceland where they will be given access to data and computation resources upon arrival to perform analyses that conform to the permissions of the Icelandic National Bioethics Committee and according to the rules of the Icelandic Data Protection Authority. The UK Biobank data were downloaded under application no. 56270. Individual level genomic and phenotypic data from the UK Biobank are available to researchers upon application (https://ukbiobank.ac.uk). Source data are provided as a Source Data file. Source data are provided with this paper.

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

## Acknowledgements

We thank the study participants as well as colleagues who contributed to data collection, sample handling, and genotyping.

## Author contributions

G.E., G.T., V.S., F.Z., H. Helgason, T.O., M.O.U., G.B., H.S., T.T., P.S., H. Holm, D.F.G., and K.S. designed the study. G.E., G.T., V.T., V.S., G.S., A.H., S.G., U.S. and H. Holm. carried out case ascertainment and performed and interpreted GWAS studies contributing data to this study. G.E., G.T., F.Z., T.O., S.R., V.T., and D.F.G. performed statistical and bioinformatics analyses. G.E., G.T., V.S., F.Z., H. Helgason, T.O., S.R., V.T., M.O.U., G.S., A.S.S., H.E., H.M.A., G.A.J., A.H., S.G., U.S., H.K.A., R.B., E.S., D.O.A., E.S.B., R.P., G.B., H.S., T.T., P.S., U.T., H. Holm, D.F.G. and K.S interpreted the results and drafted the paper. All authors contributed to the final version of this paper.

## Competing interests

G.E., G.T., V.S., F.Z., H.Helgason, T.O., S.R., V.T., M.O.U., G.S., A.S.S., H.E., H.M.A., G.A.J., A.H., S.G., U.S., H.K.A., D.O.A., G.B., H.S., T.T., P.S., U.T., H.Holm, D.F.G. and K.S. are employees of deCODE Genetics/ Amgen Inc. The remaining authors declare no competing interests.
