## [Peer Review File · Nature Communications]

REVIEWER COMMENTS

Reviewer #1 (Remarks to the Author):

Thank you for the opportunity to review this work. There are lots of elements that I like about it, but some which give pause for thought.

First, to what extent is adjustment for the putative exposure a reasonable guide whether the effect is solely mediated via the exposure or not? This is a topic that I have argued about previously with George Davey Smith - he is more sceptical of this approach. Here is a helpful thread:

<https://twitter.com/MariaGlymour/status/1092113462438723585> (starts with

https://twitter.com/mendel_random/status/1090905965673746432 - George saying in bold letters not to do this).

- There are several theoretical reasons why the exposure may be the true causal risk factor, but the genetic association does not attenuate completely. However, personally I would still expect the associations to attenuate substantially in any case. These reasons include non-linearity of the exposure-outcome effect, measurement error in the exposure, non-collapsibility, and collider bias. So the statement: "Any residual association between the BMI-GRSs and diseases after adjusting for measurements of adult BMI could be due to pleiotropy, confounding, or information not captured with few adult BMI measurements." is incomplete. It would be good to show awareness of these other potential reasons.

- There are also several reasons why the genetic association may attenuate even if the exposure is not the true risk factor - it may be correlated with the true risk factor, or there may be mediation, but incomplete mediation. Hence, it is good to focus on the risk factor being obesity in a broad sense rather than BMI in a narrow sense (as the authors do).

So personally, I think this approach is reasonable, but it would be good to be more aware of the potential limitations.

Second, to what extent is removing outliers from the analysis a reasonable approach? I hesitate here, as there is a potential for this to be a self-fulfilling prophesy. By removing outliers, could the authors have removed legitimate non-BMI pathways by which obesity may operate? But they could also have removed pleiotropic pathways that we don't want to assess here. My concern is that these outlying may represent a specific obesity-related mechanism that is not related to all BMI-related variants and does not operate via BMI, but is a legitimate mechanism, and by excluding these variants, they don't observe this pathway from obesity to disease. We (and others) have previously shown that the BMI-related variants having negative associations with T2D relate to birthweight (i.e. being a large person across lifecourse rather than being an obese person during adulthood) and do not relate to inflammation

(<https://journals.plos.org/plosgenetics/article?id=10.1371/journal.pgen.1009975>). Others have called this "metabolically healthy obesity". By removing these variants, findings potentially relate to metabolically unhealthy obesity and not to metabolically healthy obesity. On reflection, I think the authors approach (main analysis based on non-outlier variants, secondary analysis including outlier variants) is reasonable, but I would like a little more signposting that the main analyses

exclude outlier variants and consideration of the degree of attenuation with the full score in main tables/figures.

Minor points:

Would appreciate a little more care about distinguishing in the text between associations and effects: we observe associations, these provide us with evidence on the underlying mechanisms, which are effects. For example, in the Abstract: "We found that the effects of the BMI genetic risk score (GRS) associations with disease were fully attenuated..." - these are genetic associations, not effects.

"The T2D association is fully erased when conditioned on measured BMI, both in the analysis with the BMI-imGRS and the BMI-PRS, but we observe a residual association with T2D in the BMI-GRS association. These results highlight the importance of removing outlier sequence variants..." - this seems odd - it is the one analysis that removed outliers that did not see perfect attenuation. This is the opposite of expectation, no?

Figure 4: I'm not sure if this is a main figure. The point is whether the attenuation varies between variant choices when adjusting for BMI, not whether the unadjusted associations vary. Could this be better with "Proportion mediated" on the x-axis? Then we would see to what extent the attenuation varies for different choices of genetic score. Also, it'd be good to have more intuitive names / labels than "BMI-GRS", "BMI-imGRS" and "BMI-PRS" (such as "conservative score", "conservative score including outliers", "liberal score").

These comments are all suggestions for further thought, not requirements. This is a very interesting paper and the results are already clear - hopefully my comments will help to make the message clearer - please do object if you feel that the suggestions would not help the overall paper. I'd be interested in writing a supportive commentary on some of these points.

Reviewer #2 (Remarks to the Author):

This is an interesting paper that examines the mediation effect of the BMI phenotype on genetically determined effects of BMI on multiple health outcomes. The authors generated a BMI genetic risk score (GRS) to test the attenuation effect of this score on 10 outcomes after adjusting for the phenotype BMI. They demonstrated that the associations of the BMI-GRS with diseases are largely attenuated when conditioned on measured BMI, and they claim that its effects on the outcomes are mediated directly through BMI itself. However, the mediation effect and the residual effects could be explained by other mechanisms or confounding.

Major comments:

The paper uses 'obesity' to describe findings predominantly centered around 'BMI'. Technically, obesity and BMI represent different concepts. BMI is a measurement of body size and therefore a more precise definition of the research question should be provided.

The methodological approach is interesting but puzzling. Firstly, the study design and the fact that they are in effect doing one sample MR should be better explained perhaps with a schematic figure. Most importantly, conditioning on BMI, introduces collider bias. Although a sensitivity analysis of three known potential confounders was considered, there remains possibility of many other confounders. This is a major limitation of this approach. At least a thorough discussion of these biases need to be discussed.

Considering the potential for reverse causation, did the authors consider additional filters, such as Steiger filtering, be applied before constructing the GRS for BMI?

The caption for Figure 2 mentions a 'bidirectional relationship'. Does this refer to causal estimates that are both larger and smaller than zero for sets of instrumental variables (IVs)? If so, the plot does not clearly represent that outliers are mainly the cause, as removing outliers still retains bidirectional causal estimates.

'BMI were adjusted by regressing out the effect of sex and age at measurement using a generalized additive model with splines from the mgcv package²⁷. Multiple measurements were averaged after the adjustment process. The residuals from each sex were then inverse normal transformed.' Why adjust for sex first and then separately apply inverse normal transformations to the residuals for each gender? Combining both genders after these separate transformations does not ensure that the combined set of residuals will follow a normal distribution. Moreover, the transformation step differs for UK Biobank data, where gender groups are separated for all steps.

Reviewer #3 (Remarks to the Author):

I would like to congratulate the authors on an interesting study using a large sample from two different cohorts. I think the research questions addresses an important question and is something that many researchers in the field have been thinking about over the past few years.

However, I have some major concerns regarding the mediation analyses and covariate adjustments. Traditionally, a full mediation analysis requires researchers to draw a mediation diagram of the exposure, mediator and outcome, including the covariates relevant to the different sides of the mediation triangle. Once this is specified, researchers can estimate all paths, and use the produced parameters to estimate the direct effects, indirect effect and proportion mediated. This is usually achieved with a (generalised) structural mediation model, or with causal inference based g-methods. The current approach used by the authors, described as: "In the mediation analysis, we simply added the measured BMI as a covariate to the model", is not sophisticated enough to answer this quite nuanced research question. Similar issues arise in the sensitivity analyses, where potential covariates are just "added" to the regression model. However, these

covariates are not a confounders on the exposure to outcome, or exposure to mediator paths, but potentially confounders of the mediator to outcome paths. These misspecification might lead to biased results.

In light of these limitations, my recommendation would be to draw a full directional acyclic graph (DAG) including the exposure, mediator, outcomes and covariates for each model, and estimate the appropriate estimands using SEM or a causal inference based method.

An example of the latter can be found in this paper: Abdulkadir et al (2020). Polygenic Score for Body Mass Index Is Associated with Disordered Eating in a General Population Cohort. *Journal of clinical medicine*, 9(4), 1187. <https://doi.org/10.3390/jcm9041187>

In section 2.5.4. Exploratory Causal Mediation Analysis.

Thank you

Reviewer #4 (Remarks to the Author):

Point by point response

We would like to thank the reviewers for their thorough and insightful feedback on our manuscript.

We want to highlight a few major changes:

- 1) Abstract shortened to less than 150 words.
- 2) Subheadings added to results section.
- 3) Title changed.
- 4) BMI-imGRS renamed to BMI-GRS-with-Outliers
- 5) Supplementary tables are all added to the supplementary document, except for the markers used for the score, which are in a supplementary data file. This has caused some reordering of supplementary table numbers.

Below we address each point of the reviewers.

Reviewer 1 (R1)

R1 Comment 1: Thank you for the opportunity to review this work. There are lots of elements that I like about it, but some which give pause for thought. First, to what extent is adjustment for the putative exposure a reasonable guide whether the effect is solely mediated via the exposure or not? This is a topic that I have argued about previously with George Davey Smith - he is more sceptical of this approach. Here is a helpful thread:

<https://twitter.com/MariaGlymour/status/1092113462438723585>

(starts with

https://twitter.com/mendel_random/status/1090905965673746432

- George saying in bold letters not to do this).

Answer 1: Thank you for pointing this out. We were aware of George Davey Smith's stance on this matter and as he mentions in the second twitter/X link you posted; the reviewers were asking that this would be done to „test the exclusion assumption “. Our aim is not to test the exclusion assumption, but to gain insights into an upper bound on the disease risk that could be mitigated by lowering BMI and given all the caveats, we find that the mediation analysis is a sensible way to achieve that goal, but of course including all the additional sensitivity analyses. This interpretation assumes that the effect of the BMI-GRS is mediated by BMI, this is stated in the last sentence in the introduction “*Assuming that the effect of the BMI-GRS is mediated through BMI...*”.

R1 Comment 2: There are several theoretical reasons why the exposure may be the true causal risk factor, but the genetic association does not attenuate completely. However, personally I would still expect the associations to attenuate substantially in any case. These reasons include non-linearity of the exposure-outcome effect, measurement error in the exposure, non-collapsibility, and collider bias. So the statement: "Any residual association between the BMI-GRSs and diseases after adjusting for measurements of adult BMI could be due to pleiotropy, confounding, or information not captured with few adult BMI measurements." is incomplete. It would be good to show awareness of these other potential reasons.

Answer 2: You are correct, these are nuances that we should have mentioned more clearly. We have made the following amends to the manuscript. In the abstract we added mentions of these three

things “*measurement error, non-linear relationship, non-collapsibility, or confounding.*”. The following points document where we touch upon these matters in the manuscript and the supplementary note.

- 1) In the second paragraph of the discussion, we acknowledge the measurement error, although we speculate based on lifetime BMI correlations and the fact that the error is low relative to the measured values, that it is less likely to be the culprit of imperfect attenuation of the associations in the mediation analysis.
- 2) In the second to last paragraph of the results, we touch upon the topic of nonlinearity, as it likely explains the difference between the phenotype and GRS associations for heart failure. There is a brief discussion about this in the supplementary note, where we show how the ORs change upon exclusion of upper and lower quartiles of BMI for the phenotype association with the outcomes. The ORs are largely consistent, with some exceptions, e.g., for heart failure and atrial fibrillation. We feel that adding the additional remark in the abstract is justified, since we do not discuss nonlinearity specifically in the context of mediation.
- 3) We had not touched upon the issue of non-collapsibility directly. We do however perform the mediation analysis stratified by sex, a potential confounder, which yields largely consistent results, except for MI, which we highlight in the manuscript. The other adjustment covariates are year of birth and genetic principal components, which are standard in genetic risk score associations. We feel that it is justified to mention non-collapsibility in the abstract for completeness.
- 4) We address the potential influence of confounders by doing sensitivity analysis where we adjust for potential confounders. The supplementary note contains a technical discussion on how confounders can influence the regression coefficients in the mediation analysis. The formula for the regression coefficients for the GRS and BMI is as follows:

$$\mathbb{E}[Y|\mathbf{E}_{obs}, \mathbf{g}] =$$

$$\left(\beta_3 \cdot \left(1 - \frac{\sigma_0^2}{\beta_1^2 + \sigma^2 + \sigma_0^2} \right) + \left(\frac{\beta_2 \beta_1}{\beta_1^2 + \sigma^2 + \sigma_0^2} \right) \right) \cdot \mathbf{E}_{obs} + \left(\frac{\beta_3 \beta_0 \sigma_0^2 - \beta_2 \beta_1 \beta_0}{\beta_1^2 + \sigma^2 + \sigma_0^2} \right) \cdot \mathbf{g}$$

β_1 and β_2 are the parameters that stem from the confounders (see the graphical representation in the figure below). We argue in the supplementary note that the confounders are more likely to cause more attenuation, which supports further our interpretation that the proportion mediated can be interpreted as an upper bound of disease risk that can be mitigated by lowering BMI. The effect of confounders in this case is likely small, as for the other confounders that we tried adjusting for did not impact the results.

R1 Comment 3: There are also several reasons why the genetic association may attenuate even if the exposure is not the true risk factor - it may be correlated with the true risk factor, or there may be mediation, but incomplete mediation. Hence, it is good to focus on the risk factor being obesity in a broad sense rather than BMI in a narrow sense (as the authors do).

Answer 3: We agree up to a point, in that for most diseases, BMI is not the true risk factor but rather the accumulation of fat that causes disease, or a process that leads to this accumulation. Although for some diseases, such as osteoarthritis and heart failure, body mass itself may indeed be the causal risk factor. However, we note that the correlation between BMI and disease risk is present well below any reasonable definition of obesity. In any case, BMI is the only marker of fat accumulation that is available to us on a large scale, and we do not have any actual data to decipher whether BMI itself or something like the fat accumulation that it marks are truly causal. As other reviewers have commented on the difference between BMI and obesity, we have decided to change the title from “Genetic predisposition to obesity affects disease risk through obesity itself”, to “Sequence variants associated with BMI affect disease risk through BMI itself”. We have furthermore added a paragraph before the last paragraph in the discussion to address the possibility that BMI may be correlated with the true risk factor: “It is also possible that BMI itself is very correlated with the true causal factor for some of the diseases investigated, a form of vertical pleiotropy. Evidence from a clinical trial of a GLP-1 agonist as a treatment for major cardiovascular diseases demonstrates that the protective effect of treatment is observed from the start of intervention and is consistent throughout the trial period¹, yet the full effect of the treatment on weight is not observed until one year after treatment initiation². This is not consistent with the effect on cardiovascular disease being directly caused by BMI, since that would give rise to a dose-response relationship. BMI could be a consequence of an underlying causal mechanism, which could for instance be excessive food intake, adipocyte expansion or adipogenesis. This pattern of cardiovascular event incidence curves has also been observed for bariatric surgery³.”

R1 Comment 4: Second, to what extent is removing outliers from the analysis a reasonable approach? I hesitate here, as there is a potential for this to be a self-fulfilling prophesy. By removing outliers, could the authors have removed legitimate non-BMI pathways by which obesity may operate? But they could also have removed pleiotropic pathways that we don’t want to assess here. My concern is that these outlying may represent a specific obesity-related mechanism that is not related to all BMI-related variants and does not operate via BMI, but is a legitimate mechanism, and

by excluding these variants, they don't observe this pathway from obesity to disease. We (and others) have previously shown that the BMI-related variants having negative associations with T2D relate to birthweight (i.e. being a large person across lifecourse rather than being an obese person during adulthood) and do not relate to inflammation (

<https://journals.plos.org/plosgenetics/article?id=10.1371/journal.pgen.1009975>

). Others have called this "metabolically healthy obesity". By removing these variants, findings potentially relate to metabolically unhealthy obesity and not to metabolically healthy obesity. On reflection, I think the authors approach (main analysis based on non-outlier variants, secondary analysis including outlier variants) is reasonable, but I would like a little more signposting that the main analyses exclude outlier variants and consideration of the degree of attenuation with the full score in main tables/figures.

Answer 4: We read the article of Grant et al, and it is interesting to examine how the outliers we exclude fall into the clusters they define. The BMI variants used by Grant et al are from a different article⁴ than the one we use⁵. Of the 539 variants clustered by Grant et al, 107 are found in our set of 665 independent markers. The outliers are split as follows:

- 1) Cluster 1 has 52 variants and 2 are outliers.
- 2) Cluster 2 has 28 variants and 0 outliers.
- 3) Cluster 3 has 15 variants and 2 are outliers.
- 4) Cluster 4 has 8 variants and 1 is an outlier.
- 5) Cluster 5 has 4 variants and 2 are outliers.

If we also include variants that do not perfectly match, but have $LD > 0.8$, the set of overlapping variants increases from 107 to 222, the following shows the breakdown by outliers for that set:

- 1) Cluster 1 has 110 variants and 2 are outliers.
 - a. Outlier variants are both outliers for T2D and one for knee replacement.
- 2) Cluster 2 has 52 variants and 1 outlier.
 - a. Outlier for knee replacement.
- 3) Cluster 3 has 28 variants and 5 are outliers.
 - a. One outlier for T2D, one for MI, one for knee replacement, two for hip replacement, one for AF, (AF outlier variant is outlier for both AF and hip replacement).
- 4) Cluster 4 has 21 variants and 3 are outliers.
 - a. One outlier for T2D, one for MI, one for knee replacement
- 5) Cluster 5 has 11 variants and 5 are outliers.
 - a. Four are outliers for T2D, and one for MI. One of the T2D outliers is also an outlier for glucose intolerance.

It is hard to draw strong conclusions from this, albeit cluster 5 has the largest proportion of outliers, of which four of five are outliers for T2D. However, there is not an exact correspondence between the clusters of Grant et al and the variants we flagged as outliers, e.g., more than half of the variants in cluster 5 are not identified as outliers.

In our case, 25 variants are outliers for T2D and many of them were first discovered as T2D variants. We believe that reverse causation is the most likely explanation in the case of the outlier T2D variants with large effect on T2D. T2D is a heterogeneous trait, comprised of both deficiencies in insulin secretion/action, and insulin resistance. While obesity is associated with increased risk of insulin resistance, insulin deficiency can lead to weight loss. In untreated T2D cases the glucose in the blood

does not get transported to the cells, the body responds by thinking that it is starving and compensates by burning fat and muscle at a rapid pace, this causes unexplained weight loss⁶. Variants with primary effect on insulin secretion/action are not expected to associate with diabetes through increased weight gain but rather to cause weight loss through their effect on insulin. This is also known in T1D, where the telltale sign for parents is often unusual weight loss in the child. Further technical arguments are in response to comment 6.

To further highlight that the main analysis excludes outlier variants we have added the following signposts to the end of the introduction and the start of the discussion:

- 1) In the last paragraph of the introduction, added a new sentence (bold below) where we introduce the GRSs: “We utilized genetic risk scores for BMI (BMI-GRSs) and compared the association between the GRSs and the various diseases with and without adjustment for measured BMI. **The GRS in the main analysis had variants that were deemed as outliers in the BMI disease relationships excluded, as these likely violate the MR assumption that they should affect disease risk solely through the exposure.**”
- 2) The first sentence of the discussion has been changed from “Here we show that the BMI-GRS associates with various diseases, atrial fibrillation...” to “Here we show that the **conservative BMI-GRS (with outlier variants excluded)** associates with various diseases, atrial fibrillation...”.

The following are minor comments from reviewer 1:

R1 Comment 5: Would appreciate a little more care about distinguishing in the text between associations and effects: we observe associations, these provide us with evidence on the underlying mechanisms, which are effects. For example, in the Abstract: “We found that the effects of the BMI genetic risk score (GRS) associations with disease were fully attenuated...” - these are genetic associations, not effects.

Answer 5: We have changed the following sentences; changes are bold faced.

- 1) In the abstract from “*We found that the effects of the BMI genetic risk score (GRS) associations with disease were fully attenuated...*” to “**The association of the BMI genetic risk score (GRS) with disease were fully attenuated...**”.
- 2) In the abstract from “*Similarly, we observed attenuation of the effect of the BMI-GRS association with chronic kidney disease...*” to “**Similar attenuation was observed...**”
- 3) In the abstract from “*...two sexes, except for myocardial infarction where the effect of the BMI-GRS was smaller in females and was fully attenuated in females when conditioned on measured BMI but not in males.*” To “**Findings were consistent across sexes, except for myocardial infarction, where attenuation differed by sex**”.
- 4) In the abstract from “*The extent of the attenuation of the effect of BMI-GRS on diseases*” to “**The attenuation extent of BMI-GRS disease associations suggests the potential impact of reducing BMI on disease risk**”.
- 5) End of introduction from “*Assuming that the effect of the BMI-GRS is mediated through BMI, the proportion of effect attenuation in the mediation analysis provides an estimate of the upper bound of the disease risk that is potentially modifiable by lowering the BMI.*” To “**Assuming that the association of the BMI-GRS is mediated through BMI, the proportion mediated provides an estimate of the upper bound of the disease risk that is potentially modifiable by lowering the BMI.**”

- 6) The final sentence in the second paragraph of the discussion from “Assuming that the BMI-GRS confers disease risk through BMI, the proportion of its effect on disease risk that is attenuated in our mediation results provides an estimate of an upper bound of the genetic risk of the diseases that is potentially modifiable with reduction in weight.” To “Assuming that the BMI-GRS confers disease risk through BMI, **the proportion of disease risk mediated** in our mediation results provides an estimate of an upper bound of the genetic risk of the diseases that is potentially modifiable with reduction in weight.”
-

R1 Comment 6: "The T2D association is fully erased when conditioned on measured BMI, both in the analysis with the BMI-imGRS and the BMI-PRS, but we observe a residual association with T2D in the BMI-GRS association. These results highlight the importance of removing outlier sequence variants..." - this seems odd - it is the one analysis that removed outliers that did not see perfect attenuation. This is the opposite of expectation, no?

Answer 6: Since many of the outlier variants likely represent reverse causation, BMI measurements act as a collider for those variants. To exemplify this phenomenon, we have constructed a GRS exclusively from the outlier variants and redone the mediation analysis, see the figure below. Note that the ORs are for 1 SD increase in BMI. Knee replacement is the only outcome that shows results somewhat consistent with the GRS in the manuscript, hip replacement and heart failure are somewhat consistent with large CIs. Since BMI is likely mostly acting on knee replacement, through sheer mechanical force on the joint, it is likely that the underlying mechanism that creates diversity in BMI is less important in that case.

Most of the other diseases do not associate with this outlier GRS and when we condition on BMI, the effect tends away from the null, this is likely a spurious association that stems from conditioning on a collider. This is most drastic in the case of T2D where we see an association with an effect ($\log(\text{OR}) < 0$) of opposite sign to the GRS and the adjusted effect is even further away from the null.

We have added the figure for the exclusive analysis of the outliers to the supplementary note with a short description of the analysis. We reference this analysis in the last sentence of the results: “Analyzing the outlier variants exclusively further highlights these differences (Supplementary Information).”.

Score from Outliers, Associations in UKB with and w/o BMI itself
Adjusted for sex, yob and 20 PCs

R1 Comment 7: Figure 4: I'm not sure if this is a main figure. The point is whether the attenuation varies between variant choices when adjusting for BMI, not whether the unadjusted associations vary. Could this be better with "Proportion mediated" on the x-axis? Then we would see to what extent the attenuation varies for different choices of genetic score. Also, it'd be good to have more intuitive names / labels than "BMI-GRS", "BMI-imGRS" and "BMI-PRS" (such as "conservative score", "conservative score including outliers", "liberal score").

Answer 7: Figure 4 is a main figure, and we deem it more important to be able to compare the unadjusted associations before comparing the mediation results (which are in the supplementary note). We had not added the results with proportion mediated by scores, and thanks to the comment from reviewer 3, we have now reevaluated how we assess the proportion mediated. The following figures demonstrate the difference in proportion mediated by scores for the UK and Icelandic data. The values have been added to a new supplementary table 8 and proportion mediated columns from supplementary table 4 (now supplementary table 3) are removed, we will furthermore add these figures to the supplementary material. We have replaced supplementary figure 11, (see reply to reviewer 3). With the added signposting on which scores have outliers included or not (comment 4), the text is clearer on the differences between the scores. To interpret the figures more easily in isolation we have augmented the legends to include more descriptive names. We have furthermore

changed the name of “BMI-imGRS” to “BMI-GRS-with-Outliers”, since this name more clearly explains the underlying score.

Comparing proportion mediated by scores
UK data

Comparing proportion mediated by scores
Icelandic data

R1 Comment 8: These comments are all suggestions for further thought, not requirements. This is a very interesting paper and the results are already clear - hopefully my comments will help to make the message clearer - please do object if you feel that the suggestions would not help the overall paper. I'd be interested in writing a supportive commentary on some of these points.

Answer 8: We thank the reviewer for his kind words and constructive criticisms.

Reviewer 2 (R2)

R2 Comment 1: This is an interesting paper that examines the mediation effect of the BMI phenotype on genetically determined effects of BMI on multiple health outcomes. The authors generated a BMI genetic risk score (GRS) to test the attenuation effect of this score on 10 outcomes after adjusting for the phenotype BMI. They demonstrated that the associations of the BMI-GRS with diseases are largely attenuated when conditioned on measured BMI, and they claim that its effects on the outcomes are mediated directly through BMI itself. However, the mediation effect and the residual effects could be explained by other mechanisms or confounding. The paper uses 'obesity' to describe findings predominantly centered around 'BMI'. Technically, obesity and BMI represent different

concepts. BMI is a measurement of body size and therefore a more precise definition of the research question should be provided.

Answer 2: This is an accurate summary of the manuscript. Reviewer 1 touched upon the difference between BMI and obesity as well (see answer to comment 3 from reviewer 1). We changed the title of the article to further emphasize BMI instead of obesity, since BMI is our tool to make inference.

R2 Comment 2: The methodological approach is interesting but puzzling. Firstly, the study design and the fact that they are in effect doing one sample MR should be better explained perhaps with a schematic figure. Most importantly, conditioning on BMI, introduces collider bias. Although a sensitivity analysis of three known potential confounders was considered, there remains possibility of many other confounders. This is a major limitation of this approach. At least a thorough discussion of these biases need to be discussed.

Answer 2: We do not agree that this is one sample MR, since the effect estimates for the genetic risk score are based on a different sample than the sample used for the outcome association, thus there should be no bias stemming from sample overlap, this concept is presented in panel A of figure 1.

We addressed the concerns regarding confounders in answers to reviewer 1 (see responses to comments 2 and 6 of reviewer 1). We furthermore have a detailed description of how incomplete attenuation can happen in a simple structural equation model in the supplementary note, where we quantify theoretically the impact of confounders introduced via collider bias. We reference our supplementary note in the second paragraph of the discussion *“Confounders can also influence the effect of the BMI-GRS when we condition on measured BMI, but it is likely that they would bias the effect further from the unadjusted effect towards zero (Supplementary Note).”*. It is not possible to exhaustively test all confounders, and we feel that the three confounders that we test, (along with T2D diagnosis), represent a reasonable set of closely related variables to investigate in a sensitivity analysis.

R2 Comment 4: Considering the potential for reverse causation, did the authors consider additional filters, such as Steiger filtering, be applied before constructing the GRS for BMI?

Answer 4: We considered various methods to identify outliers, but in the end chose MRPRESSO, as it is robust and is based on the empirical distribution of the variants' effects.

R2 Comment 5: The caption for Figure 2 mentions a 'bidirectional relationship'. Does this refer to causal estimates that are both larger and smaller than zero for sets of instrumental variables (IVs)? If so, the plot does not clearly represent that outliers are mainly the cause, as removing outliers still retains bidirectional causal estimates.

Answer 5: We use the terminology, bidirectional relationship, since that is the wording used in the article we reference⁷. In that article, the variants are not harmonized to have the BMI effect as positive, and the 2 axes are clearer in that kind of representation. It is true, as you point out, that some points remain with a negative effect ($\log(\text{OR})$) on T2D, after removing the outliers. This is however expected, as we are dealing with a distribution, and some variants can by chance have a negative effect, yet not necessarily significantly different from zero. Even if the true effect is positive, we could not have enough statistical power to robustly estimate it.

There are 57 of the 665 BMI variants that have a negative effect (log(OR)) on T2D, which are non-outliers. These variants do not significantly associate with T2D when we account for multiple testing with the Bonferroni threshold of $0.05/620 = 8.0 \times 10^{-5}$. The smallest p -value is 0.012 and four variants are nominally significant with a p -value under 0.05. This means that we do not observe evidence of variants associating with increased BMI and lower risk of T2D among the non-outlier variants.

R2 Comment 6: 'BMI were adjusted by regressing out the effect of sex and age at measurement using a generalized additive model with splines from the mgcv package²⁷. Multiple measurements were averaged after the adjustment process. The residuals from each sex were then inverse normal transformed.' Why adjust for sex first and then separately apply inverse normal transformations to the residuals for each gender? Combining both genders after these separate transformations does not ensure that the combined set of residuals will follow a normal distribution. Moreover, the transformation step differs for UK Biobank data, where gender groups are separated for all steps.

Answer 6: Firstly, this type of adjustment is standard for genetic associations. If we do not treat the sexes separately before applying the inverse normal transform, it is likely that differences in the distribution between the sexes will cause inflations in genetic association on the X chromosome. Secondly, combining the sexes after these transformations yields a distribution that is virtually indistinguishable from a standard normal distribution – a mixture of two standard normal distributions is a standard normal distribution. Thirdly, this was not described accurately enough for the Icelandic data, the sexes were treated separately through the whole process, the methods text has been updated from "*BMI were adjusted by regressing out the effect of sex and age at measurement using a generalized additive model with splines from the mgcv package*" to "*BMI were adjusted by regressing out the effect of age at measurement using a generalized additive model with splines from the mgcv package separately for each sex*".

Reviewer 3 (R3)

R3 Comment 1: I would like to congratulate the authors on an interesting study using a large sample from two different cohorts. I think the research questions addresses an important question and is something that many researchers in the field have been thinking about over the past few years.

However, I have some major concerns regarding the mediation analyses and covariate adjustments. Traditionally, a full mediation analysis requires researchers to draw a mediation diagram of the exposure, mediator and outcome, including the covariates relevant to the different sides of the mediation triangle. Once this is specified, researchers can estimate all paths, and use the produced parameters to estimate the direct effects, indirect effect and proportion mediated. This is usually achieved with a (generalised) structural mediation model, or with causal inference based g-methods. The current approach used by the authors, described as: "In the mediation analysis, we simply added the measured BMI as a covariate to the model", is not sophisticated enough to answer this quite nuanced research question. Similar issues arise in the sensitivity analyses, where potential covariates are just "added" to the regression model. However, these covariates are not a confounders on the exposure to outcome, or exposure to mediator paths, but potentially confounders of the mediator to outcome paths. These misspecification might lead to biased results.

In light of these limitations, my recommendation would be to draw a full directional acyclic graph (DAG) including the exposure, mediator, outcomes and covariates for each model, and estimate the appropriate estimands using SEM or a causal inference based method.

An example of the latter can be found in this paper: Abdulkadir et al (2020). Polygenic Score for Body Mass Index Is Associated with Disordered Eating in a General Population Cohort. *Journal of clinical medicine*, 9(4),1187.

<https://doi.org/10.3390/jcm9041187>

In section 2.5.4. Exploratory Causal Mediation Analysis.

Thank you

Answer 1: We went through the material in the article *Polygenic Score for Body Mass Index Is Associated with Disordered Eating in a General Population Cohort* as suggested and followed the steps performed there to redo the mediation analysis using the mediation R package (version 4.5). The following figures compare the results from the quasi-Bayes method with our bootstrap approach. In the case of incomplete mediation, the methods provide very comparable results. We had initially truncated the proportion to be in the range from 0 to 1, so in the case of full attenuation, there are differences, but they do not change our interpretation of the results and the confidence intervals overlap. We have replaced supplementary figure 11 with the results from the mediation package (see the newer figure below). We have also added the results from the mediation package run to a new supplementary table 8 and removed our previous bootstrap results from supplementary table 3. We have updated the values for the proportion mediated in the results section based on this approach. We have removed the description of the bootstrap approach in the methods text and included a description of the approach with a reference to the article "*Polygenic Score for Body Mass Index Is Associated with Disordered Eating in a General Population Cohort*" and the R mediation package.

To focus more on the results rather than the methodology, we opted to include the DAGs in the supplementary text. We have a couple of diagrams with formulas showing the impact of conditioning on the putative exposure on the regression coefficients. These formulas clearly depict the impact of incomplete information of the exposure on mediation results in general.

Comparing confidence intervals for proportion mediated by methods
UK Biobank data

Comparing confidence intervals for proportion mediated by methods
Icelandic data

Proportion of BMI-GRS Association Mediated through BMI by Population and Diseases
Upper CIs clipped at 1.5

1. Lincoff, A.M. *et al.* Semaglutide and cardiovascular outcomes in obesity without diabetes. *New England Journal of Medicine* **389**, 2221-2232 (2023).
2. Ryan, D.H. *et al.* Long-term weight loss effects of semaglutide in obesity without diabetes in the SELECT trial. *Nature medicine*, 1-9 (2024).
3. Moussa, O. *et al.* Effect of bariatric surgery on long-term cardiovascular outcomes: a nationwide nested cohort study. *European Heart Journal* **41**, 2660-2667 (2020).
4. Pulit, S.L. *et al.* Meta-analysis of genome-wide association studies for body fat distribution in 694 649 individuals of European ancestry. *Human molecular genetics* **28**, 166-174 (2019).
5. Yengo, L. *et al.* Meta-analysis of genome-wide association studies for height and body mass index in ~ 700000 individuals of European ancestry. *Human molecular genetics* **27**, 3641-3649 (2018).
6. Drivsholm, T., de Fine Olivarius, N., Nielsen, A. & Siersma, V. Symptoms, signs and complications in newly diagnosed type 2 diabetic patients, and their relationship to glycaemia, blood pressure and weight. *Diabetologia* **48**, 210-214 (2005).
7. Akiyama, M. *et al.* Genome-wide association study identifies 112 new loci for body mass index in the Japanese population. *Nature genetics* **49**, 1458-1467 (2017).

REVIEWERS' COMMENTS

Reviewer #1 (Remarks to the Author):

Thanks to the authors for their thoughtful engagement with my previous comments. Nothing more to add - I appreciate the authors' viewpoints. While I may have made different choices myself in a couple of places, the authors' choices are well-justified, and the result is an interesting manuscript that I enjoyed reading. Thank you for your work on this.

Reviewer #2 (Remarks to the Author):

The authors have provided satisfactory responses to most points. However a couple of changes would improve the manuscript:

1) The authors should include the arguments and limitations discussed in the linked Twitter/x discussion in the manuscript.

2) The AUs should clarify how they are using MR-PRESSO for reverse causation clearly.

Reviewer #3 (Remarks to the Author):

Thank you for your thorough response to the reviewers' comments. I have no further comments at this stage.

Reviewer #4 (Remarks to the Author):

Point by point response

We would like to thank the reviewers again for their thorough and insightful feedback on our manuscript. Reviewer 2 had two comments in this round of revision, these comments are addressed below:

Reviewer 2 (R2)

R2 Comment 1: The authors should include the arguments and limitations discussed in the linked Twitter/x discussion in the manuscript.

Answer 1: The dispute in the twitter thread concerns few key points.

a) George Davey Smith (GDS) advocates against testing the exclusion assumption by adjusting for the putative exposure. We refer to our answer to the first comment from reviewer 1, restated here: Our aim is not to test the exclusion assumption, but to gain insights into an upper bound on the disease risk that could be mitigated by lowering BMI and given all the caveats, we find that the mediation analysis is a sensible way to achieve that goal, but of course including all the additional sensitivity analyses. This interpretation assumes that the effect of the BMI-GRS is going through measured BMI, this is stated in the last sentence in the introduction “Assuming that the effect of the BMI-GRS is mediated through BMI...”. This is also stated in the last sentence of the second paragraph of the discussion: “Assuming that the BMI-GRS confers disease risk through BMI...”.

b) The second point that GDS raises is the collider bias. This is addressed in the answer to the second comment of reviewer 1 and in the supplementary note under the title “Mediation due to partial observation of exposure and spurious associations of BMI-GRS with outcomes through confounders”. Our observation is consistent with the comment of Maria Glymour, (who replies to GDS). In this regard we show theoretically that the collider bias is likely to be small and we also test adding a few potential confounders to see if they have an impact on the association, which they don't (see the section Methods: sensitivity analysis with potential confounders and Supplementary Figure 12).

c) The third point is nonlinearities, which are now acknowledged in the manuscript. The second last sentence in the abstract is now: “Residual effects may result from temporal BMI changes, pleiotropy, measurement error, **non-linear relationships**, non-collapsibility, or confounding”.

We have modified the second paragraph of the discussion to further emphasize the limitation in points b and c. The bold text is added in this revision:

The adult BMI measurements contain incomplete information about the total exposure of BMI. The incompleteness does not stem from measurement accuracy, but rather the limited temporal coverage of repeated measurements since BMI can change over time. Adult BMI follows a predictable path, yet a limited number of measurements cannot fully inform us about the lifetime BMI trajectory. **One of the limitations of this study is that** confounders can also influence the effect of the BMI-GRS when we condition on measured BMI, but it is likely that they would bias the effect further from the unadjusted effect towards zero (Supplementary Note). **Another limitation is that Mendelian Randomization assumes a linear relationship between the exposure and outcome, nonlinear relationships can give rise to incomplete attenuation when conditioning on BMI, the putative exposure.** Assuming that the BMI-GRS confers disease risk through BMI, the proportion of disease risk mediated in our mediation results provides an estimate of an upper bound of the genetic risk of the diseases that is potentially modifiable with reduction in weight.

R2 Comment 2: The AUs should clarify how they are using MR-PRESSO for reverse causation clearly.

Answer 2: MR-PRESSO was used to remove outliers. Further analysis of these outliers strongly suggests that many are outliers stemming from reverse causation. The place in the manuscript that mentions MR-PRESSO and reverse causation is in the section “Genetic risk scores outlier removal” in the methods section. The first lines were previously: *“Due to the existence of outliers in the associations, e.g. the TCF7L2 variant for T2D (Figure 2), we decided to filter these variants out using the MRPRESSO method³⁹. We wanted to remove potential reverse causation instruments or variants that strongly deviate from the typical association pattern. Since we do not know through which biological mechanisms the associations are working, we cannot know for certain that there are any reverse causation instruments left in the pool of variants that we are using”*. We have now changed these lines to put a more general emphasis on outliers and horizontal pleiotropy. It now reads (see also track changes): *“Due to the existence of outliers in the associations, e.g. the TCF7L2 variant for T2D (Figure 2), we wanted to remove variants that strongly deviate from the typical association pattern, which could stem from horizontal pleiotropy or reverse causation. We therefore used the MRPRESSO method³⁹ to remove outliers”*.